# Silver-catalyzed direct conversion of epoxides into cyclopropanes using *N*-triftosylhydrazones

Linxuan Li [1], Paramasivam Sivaguru [1], Dandan Wei[1], Menglin Liu[1], Qingwen Zhu[1], Shuai Dong[1], Emanuele Casali [2], Nan Li[1], Giuseppe Zanoni[2] & Xihe Bi [1,3] ✉

Epoxides, as a prominent small ring *O*-heterocyclic and the privileged pharmacophores for medicinal chemistry, have recently represented an ideal substrate for the development of single-atom replacements. The previous *O*-to-*C* replacement strategy for epoxides to date typically requires high temperatures to achieve low yields and lacks substrate range and functional group tolerance, so achieving this oxygen-carbon exchange remains a formidable challenge. Here, we report a silver-catalyzed direct conversion of epoxides into trifluoromethylcyclopropanes in a single step using trifluoromethyl *N*-triftosylhydrazones as carbene precursors, thereby achieving oxygen-carbon exchange via a tandem deoxygenation/[2 + 1] cycloaddition. The reaction shows broad tolerance of functional groups, allowing routine cheletropic olefin synthesis in a strategy for the net oxygen-carbon exchange reaction. The utility of this method is further showcased with the late-stage diversification of epoxides derived from bioactive natural products and drugs. Mechanistic experiments and DFT calculations elucidate the reaction mechanism and the origin of the chemo- and stereoselectivity.

Epoxides are prominent small-ring *O*-heterocycles found in a variety of bioactive natural products and pharmaceuticals[1–3]. Moreover, they are often used as efficient structural linkers in the polymerization process[4–6] and versatile building blocks for the synthesis of complex molecules[3,7,8]. In light of their value, the transformation and utilization of readily accessible epoxides is of utmost importance in synthetic chemistry and includes the direct conversion into other three-membered (hetero) cyclic compounds such as thiiranes[9–15], aziridines[16–20], and cyclopropanes without altering the ring structure[21–24]. Given the importance of cyclopropanes in synthetic and medicinal chemistry, the conversion of epoxides to cyclopropanes (i.e., oxygen to carbon exchange) has received renewed attention. To date, four strategies have been developed to achieve this oxygen-carbon exchange in epoxides. In 1961, Wadsworth and Emmons demonstrated the first oxygen-carbon exchange in epoxides using phosphonate carbanions as the carbon

source (Fig. 1a₁)[25–29]. Subsequently, Denney and coworkers achieved this transformation by heating epoxides with Wittig reagent at 200 °C (Fig. 1a₂)[30]. Despite these advances, existing strategies generally require a high temperature to achieve oxygen-carbon exchange with low yields while lacking substrate scope and functional group tolerance. Recently, the Hong and Cho groups independently synthesized enantioenriched cyclopropylboronates via a two-step three-component reaction of chiral epoxides with lithiated *gem*-diboronates and diphenyl chlorophosphate (Fig. 1a₃)[31,32]. Noyori realized the only example of an oxygen-carbon exchange reaction by heating with ethyl diazoacetate at 130–140 °C, where small amounts of competing C–O and C–H insertion products were also formed (Fig. 1a₄)[33,34].

To broaden the scope of this transformation and generalize the conditions for the synthesis of polyfunctionalized cyclopropanes, we here report a silver-catalyzed oxygen-carbon exchange strategy for

[1]Department of Chemistry, Northeast Normal University, Changchun 130024, China. [2]Department of Chemistry, University of Pavia, Viale Taramelli 12, 27100 Pavia, Italy. [3]State Key Laboratory of Elemento-Organic Chemistry, Nankai University, Tianjin 300071, China. ✉e-mail: bixh507@nenu.edu.cn

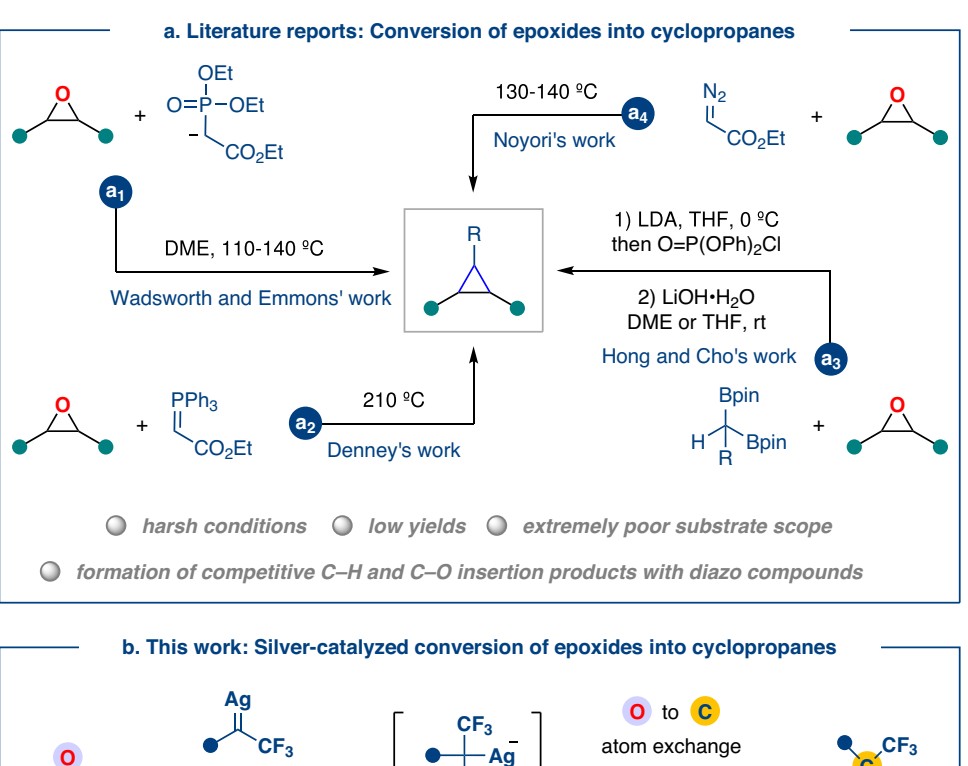

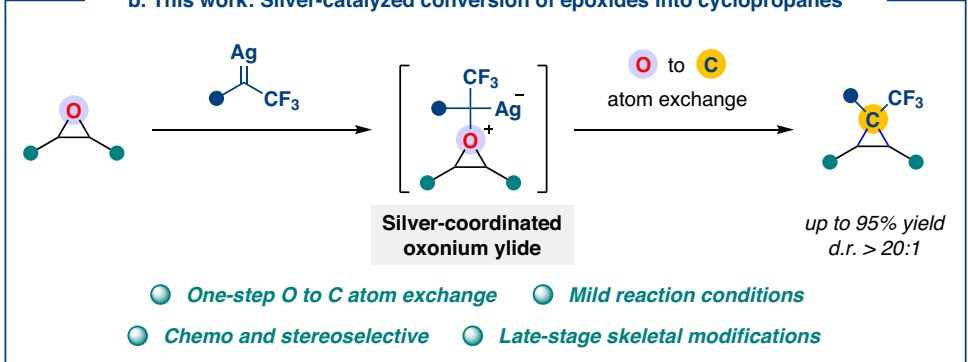

**Fig. 1 | Development of catalytic oxygen-carbon exchange reaction using carbenes. a** Literature reports: Conversion of epoxides into cyclopropanes. **b** This work: Silver-catalyzed conversion of epoxides into cyclopropanes.

the direct conversion of epoxides to cyclopropanes using fluoroalkyl *N*-triftosylhydrazones[35] under mild conditions (Fig. 1b). In contrast to existing strategies (which involve nucleophilic ring-opening of epoxides followed by ring closure with the elimination of a leaving group), this one-step process proceeds through sequential epoxides deoxygenation to furnish an alkene intermediate followed by cyclopropanation with silver carbene and the same catalyst. This protocol is operationally simple and scalable, providing trifluoromethylcyclopropanes in high yields and high diastereoselectivity with excellent functional group compatibility. Fluorinated cyclopropyl motifs are attractive synthons for the synthesis of new bioactive molecules of medicinal interest owing to the conformational rigidity of the three-membered ring combined with the unique properties imparted by the presence of fluorinated substituent[36–41].

## Results

### Optimization of the reaction conditions

At the outset, our investigations focused on the reaction of (2,3-epoxypropyl)-benzene (**1a**) and trifluoromethyl phenyl *N*-triftosylhydrazone (**2a**, 2.0 equiv.). A reaction system composed of catalytic Tp[Br3]Ag(thf) and NaH (2.5 equiv.) at 60 °C in dichloromethane (DCM) provided the desired cyclopropane **3** in 58% yield and 10:1 diastereomeric ratio (d.r.) along with 90% deoxygenation product, trifluoroacetophenone **3'** (Table 1, entry 1), without the formation of competing C–O or C–H bond insertion products[35,42–48]. Control experiments revealed that the amounts of **2a** and NaH were crucial to

improving the efficiency, and the best result (92% isolated yield of **3** and quantitative trifluoroacetophenone **3'**) was indeed obtained with 4.0 equiv. of **2a** and 4.5 equiv. of NaH (entries 1, 2 vs 3). Other silver catalysts, such as AgOTf, AgOAc, Tp[(CF3)2]Ag(thf), and Tp*Ag, resulted in diminished yields and diastereoselectivity (entries 4–7). This unusual reactivity of Tp[Br3]Ag(thf) is mainly due to the presence of a substantial steric hindrance scorpion ligand (Tp[Br3]), which not only hinders the occurrence of a competitive carbene dimerization process but also enhances non-enantioselectivity. To our surprise, no conversion of **1a** was observed when using Tp[Br3]Cu(MeCN), Pd(OAc)₂, and FeTPPCl as catalysts (entries 8–10). The use of Rh₂(OAc)₄ as catalysts results in modest efficiency in the deoxidation process, thus obtaining **3** in poor yield and selectivity (entry 11). The reaction yield and stereoselectivity could not be improved by decreasing (40 °C) or increasing (80 °C) the reaction temperature (entries 12 and 13). The substituents and positions on *N*-sulfonylhydrazones were also critical, as lower yields were observed with 4-methylphenyl- and 2,4,6-triisopropylphenyl *N*-sulfonylhydrazones (**2b** and **2c**) (entries 14 and 15). Note that the easy decomposition of *o*-trifluoromethyl *N*-triftosylhydrazones compared to other functional groups substituted *N*-sulfonylhydrazones enabled the rapid deoxidation process[35].

### Substrate scope

With the optimized conditions in hand, we explored the scope and limitations of this transformation with a range of epoxides and *N*-triftosylhydrazone (Fig. 2). We first probed the reactivity of

**Table 1 | Optimization of the reaction conditions[a]**

| Entry | 2 (equiv.) | Cat. (5 mol%) | NaH (equiv.) | Yield of 3' (%) | Yield of 3 (%) | d.r. of 3 |
|---|---|---|---|---|---|---|
| 1 | 2a (2.0) | Tp$^{Br3}$Ag(thf) | 2.5 | 90 | 58 | 10:1 |
| 2 | 2a (3.0) | Tp$^{Br3}$Ag(thf) | 3.5 | 99 | 82 | 10:1 |
| 3 | 2a (4.0) | Tp$^{Br3}$Ag(thf) | 4.5 | 99 | 95 (92)[d] | 10:1 |
| 4 | 2a (4.0) | AgOTf | 4.5 | 30 | 12 | 5:1 |
| 5 | 2a (4.0) | AgOAc | 4.5 | 43 | 32 | 7:1 |
| 6 | 2a (4.0) | Tp$^{(CF3)2}$Ag(thf) | 4.5 | 82 | 60 | 7:1 |
| 7 | 2a (4.0) | Tp*Ag | 4.5 | 60 | 22 | 7:1 |
| 8 | 2a (4.0) | Tp$^{Br3}$Cu(MeCN) | 4.5 | 56 | 0 | – |
| 9 | 2a (4.0) | Pd(OAc)$_2$ | 4.5 | 0 | 0 | – |
| 10 | 2a (4.0) | FeTPPCl | 4.5 | 8 | Trace | – |
| 11 | 2a (4.0) | Rh$_2$(OAc)$_4$ | 4.5 | 60 | 10 | 4:1 |
| 12[b] | 2a (4.0) | Tp$^{Br3}$Ag(thf) | 4.5 | 89 | 28 | 10:1 |
| 13[c] | 2a (4.0) | Tp$^{Br3}$Ag(thf) | 4.5 | 95 | 76 | 9:1 |
| 14 | 2b (4.0) | Tp$^{Br3}$Ag(thf) | 4.5 | 67 | 37 | 10:1 |
| 15 | 2c (4.0) | Tp$^{Br3}$Ag(thf) | 4.5 | 62 | 33 | 8:1 |

[a]Reaction conditions: **1** (0.15 mmol), **2** (0.6 mmol), NaH (0.68 mmol), *Cat.* (5 mol%) in DCM (4.0 mL) at 60 °C under N$_2$ for 6 h.
[b]40 °C instead of 60 °C.
[c]80 °C instead of 60 °C.
[d]Isolated yields. The yields and the diastereomeric ratio (d.r.) were determined by the relative integration of $^{19}$F NMR spectra.

various epoxides for this oxygen-carbon exchange protocol using 4-chlorophenyl trifluoromethyl *N*-triftosylhydrazone as a carbon component (Fig. 2a). Mono-substituted epoxides with methyl, *n*-butyl, *n*-decyl, benzyl, and 2-phenethyl groups readily transformed into corresponding cyclopropane products **4–8** with good to excellent yields and good diastereomeric ratio (d.r.). Dialkyl ether and alkylaryl ether-tethered epoxides were tolerated well, giving rise to cyclopropane products **9–12** in good yields and diastereoselectivities (>20:1 d.r.), except for product **9** (7:1 d.r.). Pleasingly, functional groups such as ester, chloro, and tosylate groups at the terminus of alkyl epoxides underwent smooth reaction, affording cyclopropane products **13–15** in good yields and diastereoselectivities. 3-Phenyl-2,2'-bisepoxide was transformed into biscyclopropane **16** with high diastereoselectivity albeit in lower yield under modified conditions for 24 h.

This catalytic protocol also operates effectively with 2-(phenylethynyl)oxirane to furnish cyclopropane **17** (54% yield and >20:1 d.r.), where the internal alkynyl remains intact for [2 + 1] cycloaddition with carbenes. Under standard conditions, 1,2,7,8-diepoxyoctane readily converted to *n*-hexene-substituted cyclopropane **18** (80% yield and >12:1 d.r.) but produced the expected 1,4-dicyclopropylbutane **19** (74% yield and 8:1 d.r.) with 6 equiv. of 4-chlorophenyl trifluoromethyl *N*-triftosylhydrazone. Next, an array of aryl-substituted oxiranes was subjected to the reaction system. Aryl epoxides with various substitution groups (e.g., methyl, phenyl, chloro, methoxy, and trifluoromethyl) and electronic properties were well tolerated with this

transformation, affording cyclopropanes **20–28** in high yields (62–93%) and excellent diastereoselectivities (>20:1 d.r.). Similarly, disubstituted phenyl-, naphthyl-, and 2-thienyl-substituted epoxides were converted into cyclopropanes **29–31** in good yield with >20:1 d.r. The scope of various disubstituted oxiranes was then examined. Several 2,2-disubstituted epoxides, including 2,2-dimethyl, sterically hindered 2,2-diaryl, and 2-(cyclo)alkyl-2-aryl-substituted epoxides, as well as cyclohexyl ketal, pyran, *N*-Cbz protected piperidine fused epoxides proved compatible, leading to highly substituted cyclopropanes **32–40** in high yields with d.r. as high as 16:1. Similarly, various 2,3-disubstituted epoxides (e.g., cyclopentane, 2,3-dihydro indene oxirane, 2,3-dialkyl-, 2-alkyl-3-aryl-, and 2,3-diaryl-substituted epoxides) afforded the corresponding cyclopropanes **41–45** with good to excellent yields and moderate to good d.r.

Finally, we explored the scope of fluoroalkyl *N*-triftosylhydrazones with the representative styrene oxide or 4-methoxystyrene oxide (Fig. 2b). Under optimized conditions, both electron-rich and electron-poor aryl trifluoromethyl *N*-triftosylhydrazones were well tolerated with this transformation, furnishing cyclopropane products **46–55** in high yields and excellent d.r. Similarly, disubstituted phenyl-, naphthyl-, and furyl-substituted trifluoromethyl *N*-triftosylhydrazones reacted smoothly to afford corresponding cyclopropanes **56–58** in 53–78% yield and >20:1 d.r. *N*-Triftosylhydrazones derived from trifluoromethyl vinyl/1,3-dienyl ketones were also found to be suitable carbene precursors, affording the corresponding alkenyl/dienyl

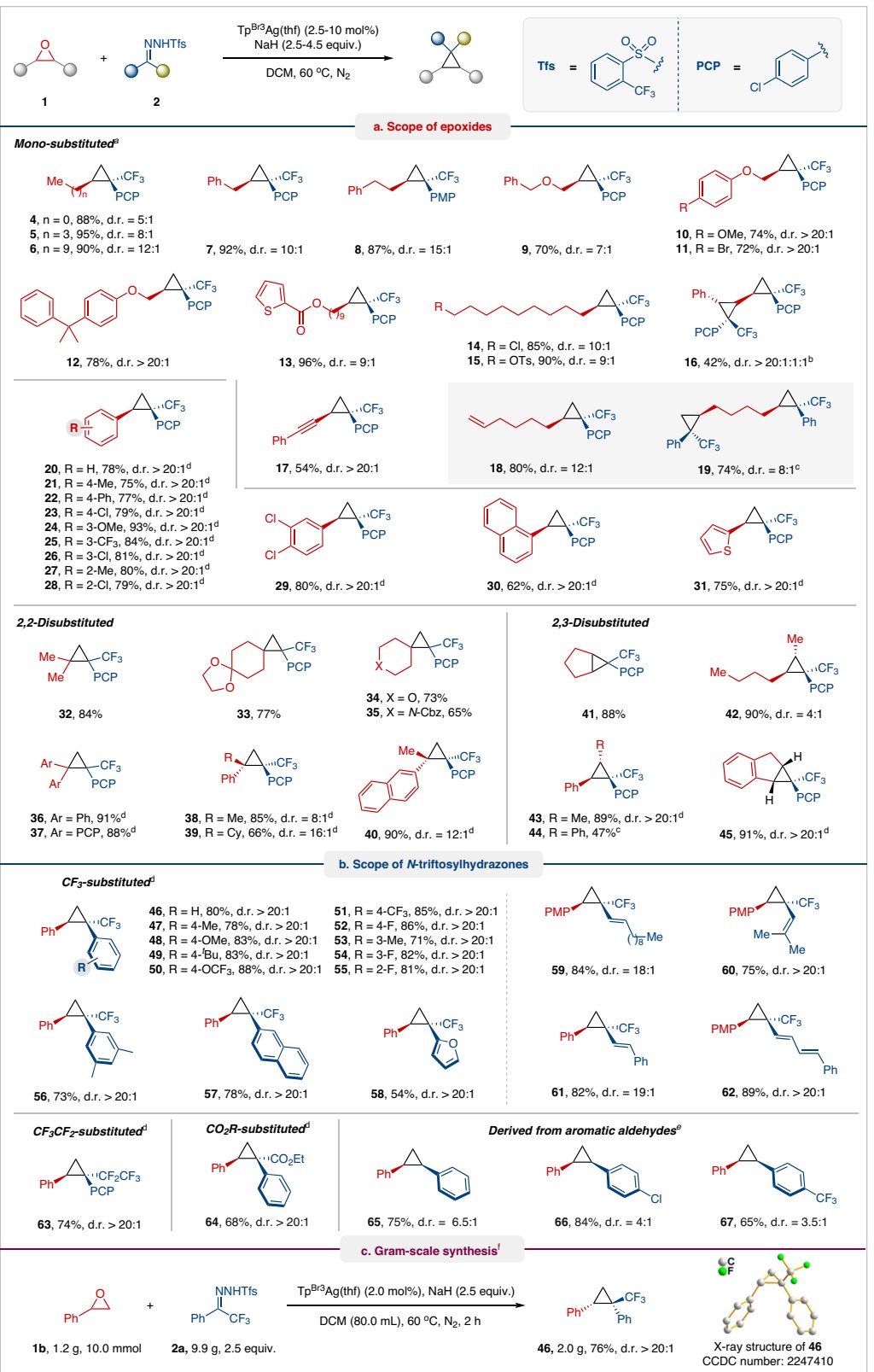

**Fig. 2 | Scope of single-atom transmutation of epoxides with fluoroalkyl *N*-triftosylhydrazones. a** Scope of epoxides. **b** Scope of *N*-triftosylhydrazones. **c** Gram-scale synthesis. Reaction conditions: [a]**1** (0.15 mmol, 1.0 equiv.), **2** (0.6 mmol, 4.0 equiv.), NaH (0.68 mmol, 4.5 equiv.) and Tp^Br3Ag(thf) (10 mol%) in DCM (4.0 mL) at 60 °C for 6–12 h. [b]Extend reaction time to 24 h. [c]Using 6.0 equiv. of **2**. [d]**1** (0.15 mmol, 1.0 equiv.), **2** (0.38 mmol, 2.5 equiv.), NaH (0.38 mmol, 2.5 equiv.) and Tp^Br3Ag(thf) (2.5–5 mol%) in DCM (4.0 mL) at 60 °C for 0.5–6 h. [e]**1** (0.15 mmol,

1.0 equiv.), **2** (0.45 mmol, 3.0 equiv.), NaH (0.45 mmol, 3.0 equiv.) and Tp^Br3Ag(thf) (5 mol%) in DCM (4.0 mL) at 60 °C for 10 h. [f]**1b** (1.2 g, 10 mmol), **2a** (9.9 g, 25 mmol), NaH (1.0 g, 25 mmol) and Tp^Br3Ag(thf) (219 mg, 2 mol%) in DCM (80.0 mL) at 60 °C for 2 h. Yields of the isolated product are given. Diastereomeric ratio (d.r.) determination by relative integration of ^19F NMR spectra. PCP, 4-chlorophenyl. PMP, 4-methoxyphenyl.

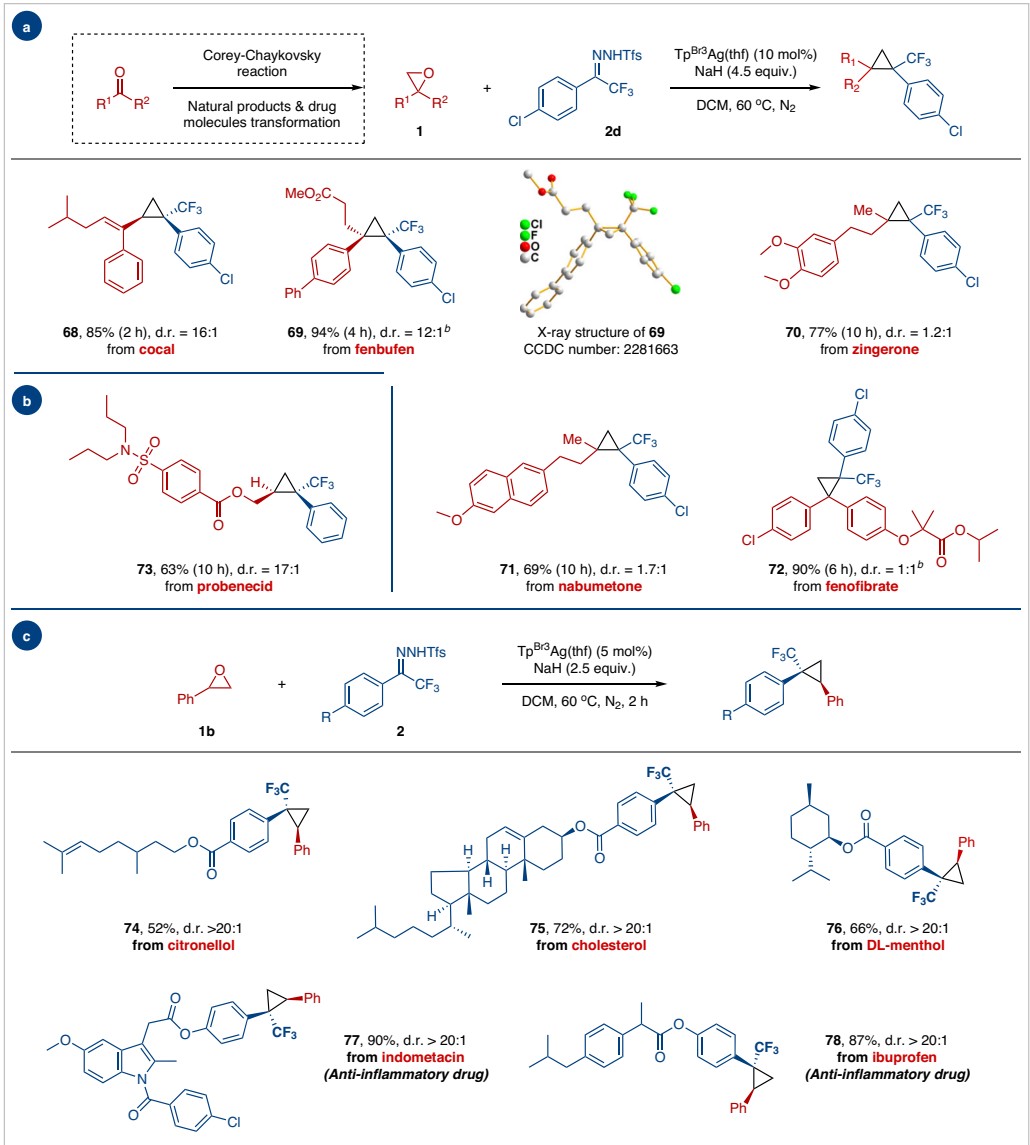

**Fig. 3 | Late-stage skeletal editing to access trifluoromethylcyclopropane containing quaternary carbon centers. a, b** Scope of epoxides derived from natural products and drugs. **c** Scope of *N*-triftosylhydrazones derived from natural products and drugs. Reaction conditions: [a]**1** (0.15 mmol, 1.0 equiv.), **2d** (0.6 mmol, 4.0 equiv.), NaH (0.68 mmol, 4.5 equiv.) and TpBr3Ag(thf) (10 mol%) in DCM (4.0 mL) at 60 °C for 2–10 h. [b]**1b** (0.15 mmol, 1.0 equiv.), **2** (0.38 mmol, 2.5 equiv.), NaH (0.38 mmol, 2.5 equiv.) and TpBr3Ag(thf) (5 mol%) in DCM (4.0 mL) at 60 °C for 2 h. Yields of the isolated product are given. Diastereomeric ratio (d.r.) determination by relative integration of ¹⁹F NMR spectra.

substituted cyclopropanes **59**–**62** in high yields (75–89%) with excellent diastereoselectivity. We also attempted to use alkyl *N*-triftosylhydrazones as carbene precursors (For details, please see Supplementary Fig. 1b), but no cyclopropane product was observed owing to the in situ generated alkyl trifluoromethyl silver carbene more prone to undergo β-hydrogen elimination to yield trifluoromethyl substituted olefins. Even methyl trifluoromethyl *N*-triftosylhydrazone also failed to yield the cyclopropanation product, likely due to the low reactivity of alkyl trifluoromethyl silver carbene. Interestingly, *N*-triftosylhydrazone derived from pentafluoroethyl phenyl ketone was successfully converted into pentafluoroethyl cyclopropane **63** in 74% yield with a d.r. of >20:1, demonstrating the capability of the present work in divergent synthesis. This protocol also successfully extended to *N*-triftosylhydrazones derived from ethyl phenylglyoxylate with good yield and excellent selectivity. Aside from trifluoromethyl ketones derived *N*-triftosylhydrazones, those derived from aromatic aldehydes performed well to afford desired 1,2-diaryl cyclopropanes **65**–**67** in

moderate diastereoselectivities. To demonstrate the robustness and scalability of this protocol, we performed a 10.0 mmol scale transformation to prepare cyclopropane **46** without lowering the yield or diastereoselectivity of the reaction (Fig. 2c). The relative configuration of **46** was confirmed with X-ray crystallography.

Given the importance of trifluoromethylcyclopropane motifs in bioactive molecules, epoxides derived from natural products and drugs were synthesized to showcase the versatility of the protocol (Fig. 3). Under standard conditions, epoxides derived from cocal, fenbufen, zingerone, nabumetone, and fenofibrate readily transformed into corresponding trifluoromethylcyclopropane products **68**–**72** in moderate to high yield and diastereoselectivities. Similarly, the trifluoromethylcyclopropane functionality could be successfully installed onto the sulfonamide anti-gout drug probenecid **73** (63% yield and 17:1 d.r.). We attempted to use allose-derived epoxides as substrates for the reaction because the steric hindrance of the trisubstituted olefin intermediates prevented further conversion of

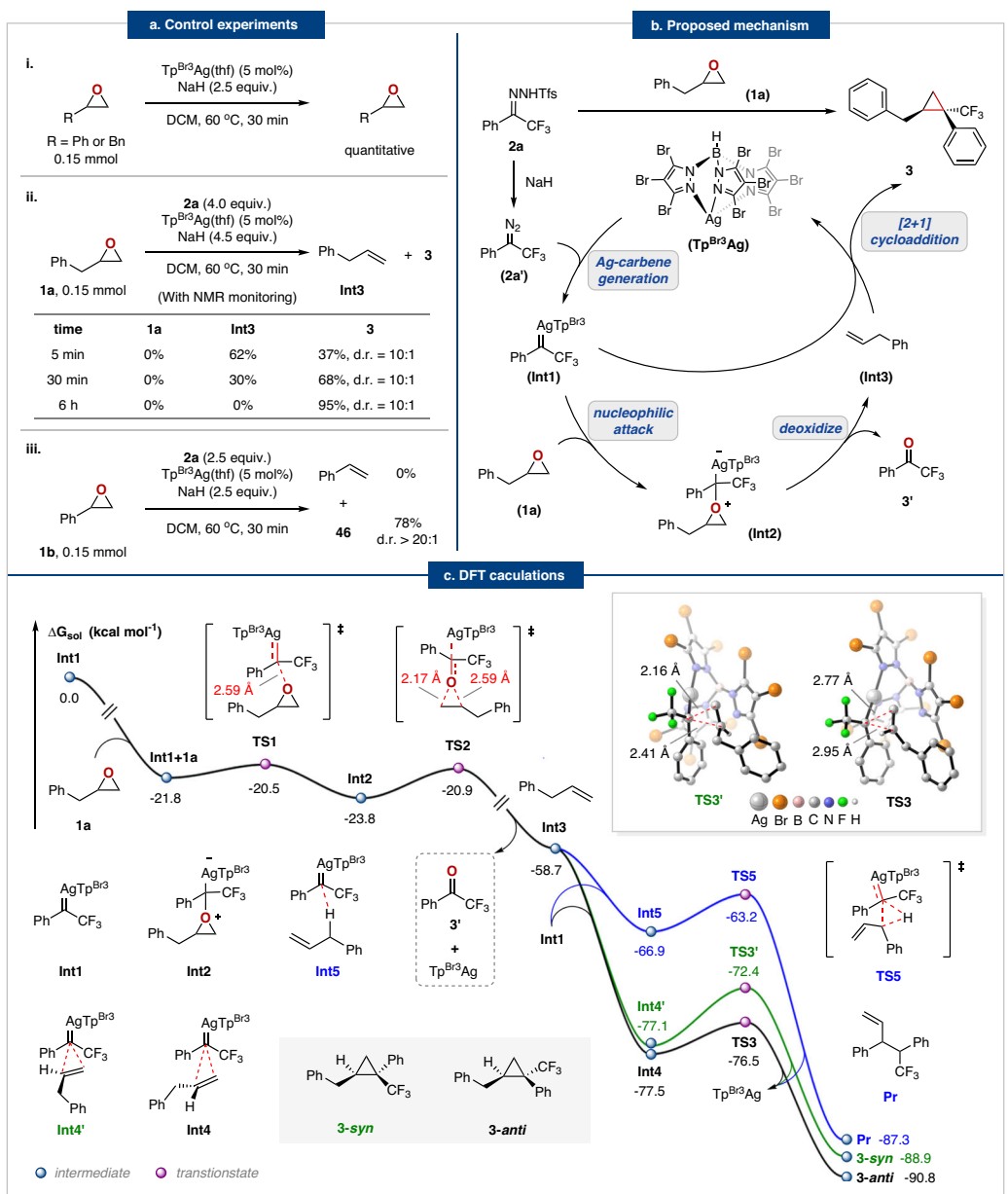

**Fig. 4 | Mechanism investigation. a** Control experiments. **b** Proposed mechanism. **c** Computed fragmentation/[2 + 1] cycloaddition mechanism. Energies are given in kcal mol⁻¹ and most H atoms in 3D structures are omitted for clarity.

cyclopropane. The derivatives of scopolamine, due to their alkalinity, result in catalyst deactivation and failure to obtain the corresponding cyclopropane product (For details, please see Supplementary Fig. 1b). The anti-gout drug probenecid was used to treat severe acute respiratory syndrome-coronavirus-2 (SARS-CoV-2) infection[49,50]. Moreover, *N*-triftosylhydrazones derived from bioactive natural products (e.g., citronellol, cholesterol, and DL-menthol) and nonsteroidal anti-inflammatory drugs (e.g., indometacin and ibuprofen) effectively underwent this oxygen-carbon exchange reaction to afford cyclopropane products **74**–**78** in high yields with excellent stereocontrol.

## Mechanistic investigations

We next conducted control experiments to gain some insights into the reaction mechanism (Fig. 4a). First, subjecting (2,3-epoxypropyl)-benzene or styrene oxide to standard conditions without *N*-triftosylhydrazones leads to no reaction and recovered epoxides in quantitative yields (Fig. 4a, eq. i), indicating the high stability of epoxides in basic conditions and *N*-triftosylhydrazone is necessary for deoxygenation

process. However, the model reaction in the presence of *N*-triftosylhydrazones, both (2,3-epoxypropyl)-benzene and styrene oxide disappeared within 5 mins and produced olefin intermediate and cyclopropane, suggesting that the in situ-generated silver carbene may induce this sequential deoxygenation and cycloaddition process. When the reaction lasted for 30 mins, (2,3-epoxypropyl)-benzene produced allylbenzene (30%) along with the expected cyclopropane product **3** (68%), whereas styrene oxide gave the cyclopropane product **46** in 78% yield (Fig. 4a, eq. ii and iii), owing to the higher reactivity of aryl olefins toward cyclopropanation than alkyl olefins. After 6 hours of reaction, allylbenzene was completely consumed and converted to cyclopropane product **3** (92%, d.r. = 10:1) (Fig. 4a, eq. ii). Based on the control experiments, we proposed a reasonable deoxygenation and cycloaddition mechanism in Fig. 4b.

To gain a better understanding of the reaction mechanism and the specific interactions responsible for the experimentally observed stereoselectivity, we performed systematic density functional theory (DFT) calculations for the reaction between trifluoromethyl phenyl

diazomethane **2a'** and (2,3-epoxypropyl)-benzene at the B3LYP/6-31 G(d,p)-GD3(BJ)-SDD (Ag,Br) level using PCM (CH$_2$Cl$_2$) solvent model. The calculations indicated that the most favorable pathway was net oxygen-carbon exchange via cycloaddition of the in situ generated olefin from deoxygenation of epoxides with excess silver carbene. Figure 4c depicts the calculated free energy profiles for the catalytic cycle. First, the dinitrogen extrusion from **2a'** occurs in the presence of an Ag catalyst via the transition state **TS1'** ($\Delta G^{\neq} = 8.6$ kcal mol$^{-1}$) to form an electrophilic silver carbene **Int1** (For details, please see Supplementary Fig. 2a). Then the **Int1** either binds to (2,3-epoxypropyl)-benzene **1a** to form **Int2** (**TS1**, $\Delta G^{\neq} = 1.3$ kcal mol$^{-1}$) or reacts with excess **2a'** to provide the **Pr'** via **TS4** ($\Delta G^{\neq} = 4.7$ kcal mol$^{-1}$) (For details, please see Supplementary Fig. 2b), which is consistent with the experimental results. However, the calculations suggest that the formation of **Int2** is preferable due to the low energy barrier required. As a result, **Int2** readily undergoes intramolecular ring-opening isomerization via **TS2** ($\Delta G^{\neq} = 2.9$ kcal mol$^{-1}$) to form trifluoromethyl acetophenone **3a** and allylbenzene **Int3**. Finally, the concerted [2 + 1] cycloaddition of **Int3** with the excess silver carbene **Int1** yields the cyclopropane product **3-syn** ($\Delta G^{\neq} = 4.7$ kcal mol$^{-1}$) and **3-anti** ($\Delta G^{\neq} = 1.0$ kcal mol$^{-1}$). Note that competitive allylic C−H bond insertion might occur to some extent due to the presence of allylic C−H bond[51]. Our calculations demonstrated that the C−H bond insertion reaction at the allylic position occurs via the three-membered ring transition state formed by **Int1** and **Int3** (**TS5**, $\Delta G^{\neq} = 3.7$ kcal mol$^{-1}$). When compared to the C−H insertion process, the [2 + 1] cycloaddition is both thermodynamically and kinetically favored and thus produces cyclopropane products **3-anti** and **3-syn**.

The cyclopropane (**3-anti** and **3-syn**) formation transition states **TS3** and **TS3'** are shown in Fig. 5. According to our calculations, the number of C−H⋯F hydrogen bonding interactions in **TS3'** is more than that of **TS3**, resulting in a sterically more hindered configuration of **TS3'** (Fig. 5). The color-filled reduced density gradient (RDG) analysis[52,53] for

the cyclopropanation process demonstrated that **TS3** has a stronger non-covalent interaction of mutual attraction, while a stronger steric hindrance exists in **TS3'**. Thus, the formation of **3-anti** is highly favorable over **3-syn**.

## Discussion

In summary, we have demonstrated the use of fluoroalkyl *N*-triftosylhydrazones as competent one-carbon atom reagents for the one-step conversion of epoxides to cyclopropanes under mild conditions. The oxygen-carbon exchange method described herein provides rapid access to diverse valuable fluoroalkyl cyclopropanes in high yields and diastereoselectivity and shows good functional group compatibility. This method can be successfully extended to the late-stage functionalization and diversification of pharmaceutically relevant bioactive natural products and drugs. Examination of the reaction mechanism by DFT supports the ylide fragmentation-cycloaddition sequence initiated by fluoroalkyl carbene in an unusual concerted process. This report represents an important example of the emerging concept of "single-atom skeletal editing".

## Methods

### General procedure for the synthesis of fluoroalkyl *N*-triftosylhydrazones

Under air conditions, fluoroalkyl ketone (11 mmol, 1.1 equiv.) and 20.0 mL ethyl acetate were added in an oven-dried 50.0 mL round-bottomed flask and then TfsNHNH$_2$ (10 mmol, 1.0 equiv.) was added and the reaction mixture was stirred at 40 °C. Upon complete dissolution, 2.0 mL of boron trifluoride ether (48 wt% BF$_3$) was added. After complete consumption of the starting material (as evidenced by TLC), saturated sodium chloride solution was added and then extracted with ethyl acetate (20.0 mL × 3). The combined organic layers were dried over anhydrous Na$_2$SO$_4$ and concentrated under reduced pressure. The crude product was purified by silica gel column chromatography (petroleum ether/ethyl acetate = 10/1) to obtain the product with a white solid.

### General procedure oxygen-carbon exchange reaction of alkyl substituted epoxides

In the glove box, *N*-sulfonylhydrazones derived from aryl trifluoromethyl ketone (0.60 mmol, 4.0 equiv.), NaH (0.68 mmol, 4.5 equiv.), and 4.0 mL of DCM were added to the dry-sealed tube. Then the alkyl-substituted epoxides (0.15 mmol) and Tp$^{Br3}$Ag(thf) (10 mol%) were added under stirring conditions. The resulting mixture was sealed and heated to 60 °C. After the reaction was completed and cooled to room temperature, the reaction mixture was filtered through diatomaceous earth under reduced pressure and the filter pad was washed with DCM (5.0 mL × 3). The combined residue was then concentrated under reduced pressure and the crude residue was purified by flash silica gel column chromatography (petroleum ether as eluent) to give cyclopropanation products.

### General procedure oxygen-carbon exchange reaction of aryl-substituted epoxides

In the glove box, *N*-sulfonylhydrazones derived from aryl trifluoromethyl ketone (0.38 mmol, 2.5 equiv.), NaH (0.38 mmol, 2.5 equiv.), and 4.0 mL of DCM were added to the dry-sealed tube. Then the aryl-substituted epoxides (0.15 mmol) and Tp$^{Br3}$Ag(thf) (2.5–5 mol%) were added under stirring conditions. The resulting mixture was sealed and heated to 60 °C. After the reaction was completed and cooled to room temperature, the reaction mixture was filtered through diatomaceous earth under reduced pressure and the filter pad was washed with DCM (5.0 mL × 3). The combined residue was then concentrated under reduced pressure and the crude residue was purified by flash silica gel column chromatography (petroleum ether as eluent) to give cyclopropanation products.

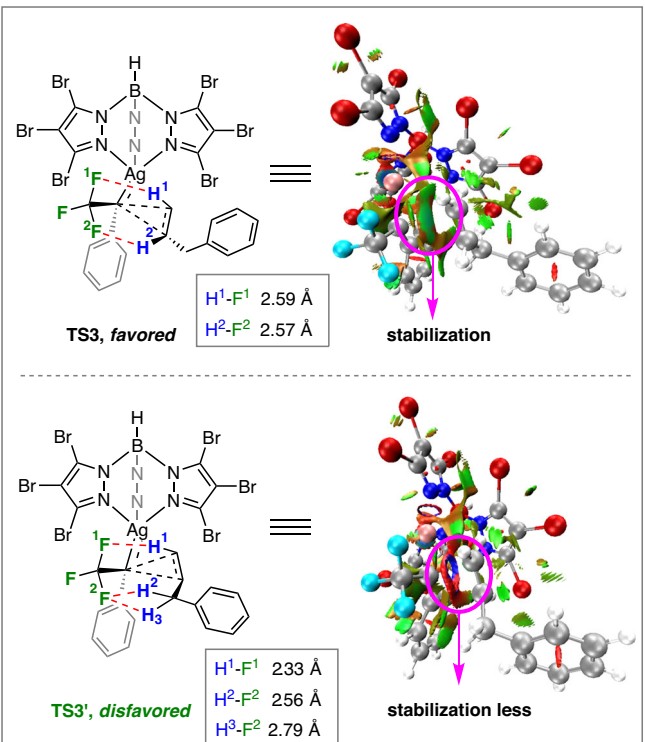

| | |
|---|---|
| H$^1$-F$^1$ | 2.59 Å |
| H$^2$-F$^2$ | 2.57 Å |

**TS3, favored**   **stabilization**

| | |
|---|---|
| H$^1$-F$^1$ | 233 Å |
| H$^2$-F$^2$ | 256 Å |
| H$^3$-F$^2$ | 2.79 Å |

**TS3', disfavored**   **stabilization less**

**Fig. 5 | The number of C−H⋯F hydrogen bonding interactions (left).** color-filled reduced density gradient (RDG) isosurface for **TS3** and **TS3'** (isovalue set to 0.5): (blue) areas of attraction (covalent bonding); (green) vdW interaction; (red) areas of repulsion (steric and ring effects) (right).

**General procedure oxygen-carbon exchange reaction of *N*-trif-tosylhydrazones derived from aryl aldehyde**

In the glove box, *N*-triftosylhydrazones derived from aryl aldehyde (0.45 mmol, 3.0 equiv.), NaH (0.45 mmol, 3.0 equiv.), and 4.0 mL of DCM were added to the dry-sealed tube. Then the aryl-substituted epoxides (0.15 mmol) and Tp^{Br3}Ag(thf) (5 mol%) were added under stirring conditions. The resulting mixture was sealed and heated to 60 °C. After the reaction was completed and cooled to room temperature, the reaction mixture was filtered through diatomaceous earth under reduced pressure and the filter pad was washed with DCM (5.0 mL × 3). The combined residue was then concentrated under reduced pressure and the crude residue was purified by flash silica gel column chromatography (petroleum ether as eluent) to give cyclopropanation products.

## Data availability

The data reported in this paper are available in the main text or the Supplementary Information. The Cartesian coordinates of all optimized structures are available from the Supplementary Data 1. Crystallographic data for the structures reported in this Article have been deposited at the Cambridge Crystallographic Data Centre, under deposition numbers CCDC 2247410 (compound **46**), and 2281663 (compound **69**). Copies of the data can be obtained free of charge via https://www.ccdc.cam.ac.uk/structures/. All other data are available in the main text or the Supplementary Information. All data are available from the corresponding author upon request.

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

## Acknowledgements
We acknowledge the National Natural Science Foundation of China (NSFC) (grant No. 22331004 to X.B.) for financial support.

## Author contributions
L.L., M.L., Q.Z., S.D., and N.L. performed the experimental investigations and D.W. carried out the theoretical calculations. L.L. and X.B. conceived the concept, designed the project, analyzed the data, and together with E.C., P.S., and G.Z. discussed the results and prepared this manuscript.

## Competing interests
The authors declare no competing interests.
