## [Peer Review File · Nature Communications]

REVIEWER COMMENTS

Reviewer #1 (Remarks to the Author):

The manuscript submitted by Professor Xihe Bi and co-workers presents a synthetic method of cyclopropanes with CF₃ group from epoxides. The developed reaction efficiently proceeds under mild conditions relative to the previous methods. The synthetic value of the unique transformation using Ag-carbene species was demonstrated by the derivatization of bioactive natural products and drug-like molecules with an array of functionalities.

The mechanistic analysis based on the DFT computation is also informative for potential readers. I, therefore, would like to recommend the publication of this paper in Nature Communications. I'd like to suggest some corrections and comments before publication.

1) About the title, the developed method fundamentally provides the cyclopropanes possessing CF₃ group. Thus, a more concrete title might be appropriate, i.e., "Late-stage skeletal transmutation of epoxides into cyclopropanes with trifluoromethyl group".

2) In Figure 2, please spell out the "PCP" if it does not.

3) In Figure 4, regarding the reaction mechanism based on the DFT computation, I think that the most impressive elementary reaction is the concerted process from Int2 into Int3 via TS2. It would be appreciated if the authors also consider the stepwise pathway, like the attachments file.

Reviewer #2 (Remarks to the Author):

This paper describes a new cyclopropanation method via deoxygenation of epoxides under relatively mild conditions. With the careful choice of silver catalysts and sulfonylhydrazones as the carbene precursor, the direct transformation of a range of substituted epoxides was successfully achieved with favorable diastereoselectivity. Some control experiments along with DFT calculations provide mechanistic insights into the consecutive process which involves generation of alkene intermediates and subsequent [2+1] cycloaddition. Regarding the possible formation of ketones as the side product in the deoxygenation step, it's advisable to confirm the yield of trifluoromethyl ketone in the initial screening experiments of Table 1. Although other experimental results including synthetic applications are adequately provided, I am concerned about some weaknesses of this manuscript; (1) a limited scope of hydrazones accessible mainly from aromatic trifluoromethyl ketones, as well as (2) unachieved enantioselective applications. In sum, this is an excellent synthetic method contribution within its specific, demonstrated scope, but I feel it is just one step short of being published in a prestigious journal, Nat. Commun.

Minor issues:

(1) To corroborate the cyclopropanation pathway through the free alkene intermediates, it would be better to check a complete loss of optical purity in the reaction with nonracemic epoxides.

(2) For the products 16 and 19 containing two cyclopropane units, relative stereochemical ratios other than the diastereomeric ratios due to 1,2-substituents on the cyclopropane need to be specified.

(3) In Figure 4B, I advise to renumber the product 3a, trifluoromethyl acetophenone, which may confound with the cyclopropane product 3.

(4) Throughout the Supplementary Information; “trimethylsulfoxide iodide” should be replaced with “trimethylsulfoxonium iodide”.

(5) Typos in the Supplementary Information:

P. 15, L .7; hydrotris(3,5-dimethylpyrazol-1-yl)borate.

P. 18; DFT Calculations.

Reviewer #3 (Remarks to the Author):

In this manuscript by Bi and coworkers, the transformation of epoxides into cyclopropanes in one pot by the action of N-trifosylhydrazones as carbene precursors (and deoxygenation reagents) and a silver catalyst under basic conditions is reported. The authors demonstrate a remarkable substrate scope and functional group tolerance. The process performs equally well for some commercial drugs and other highly functionalized substrates. Based on control experiments and DFT computations, a reasonable mechanism is proposed. It consists of sequential epoxide deoxygenation with oxygen capture by the silver-carbene and cyclopropanation of the olefine intermediates to yield the products in good stereoselectivities. The entire process is termed as “skeletal editing”, a field of high topical interest. Overall, the described protocol grants access to a wide range of valuable products. However, at a second look, it is merely a one-pot process of epoxide deoxygenation (well-known) and olefin-cyclopropanation (well-known). This fact becomes further obvious because substrates containing terminal or disubstituted olefins do not appear in the substrate table. This shortcoming becomes apparent after consideration of the mechanism, but the authors failed to discuss this at an earlier stage. Thus, I am missing the conceptual novelty of the process if it would correspond to a proper “single step” skeletal editing, and I am less enthusiastic about recommending the manuscript for publication in Nat. Comm. Indeed, I would expect a less impactful reception of the same work if the terminology “one-pot” or “tandem,” more realistically reflecting the mechanism, would have been chosen. I am not an expert in organic synthesis, so I am unsure if this criticism is also mirrored in the broader community.

Beyond that, the manuscript is well written, and the synthesis, analysis, and computational work is performed at a high level. The following points should be considered in any case:

1) A more precise association of the processes in Figure 1 with the text, either by names or a)b) etc, would be helpful.

2) In substrate 8, Figure 2, why PMP?

3) In the discussion of the substrate scope, please comment on aliphatic carbene precursors.

4) Page 6, line 133: not “alkaline” but rather “basic” conditions.

5) Page 7, line 164: “thermodynamically and kinetically more stable” should be “thermodynamically and kinetically favored”

6) Page 7, line 167: Please refer to Figure 5

7) Improve the resolution of Figures 4 and 5.

Reviewer #4 (Remarks to the Author):

The manuscript by Li and coauthors describes a silver-catalyzed conversion of epoxides into cyclopropanes using N-triflylhydrazones as carbene precursors via an oxygen-carbon exchange process. The reaction proceeds through initial conversion of the epoxide into the olefin, followed by subsequent cyclopropanation of the olefin. Mechanistic experiments, along with DFT calculations, support a reaction pathway involving silver-coordinated ylide fragmentation, followed by a [2+1] cycloaddition.

Overall, the manuscript is well written, with aesthetically pleasing figures. The substrate scope was shown to be quite broad, with good functional group tolerance. However, from a medicinal chemistry perspective, additional azaheterocycles would be desired.

While the overall transformation is quite interesting, this reviewer is unsure of the benefit of converting the carbonyl group to the epoxide vs directly to the olefin (which is formed in situ in the reaction sequence) for cyclopropanation. In cases where the epoxide already exists, this method would be great, however, this is likely more niche in nature. For this reason, this reviewer suggests publication in a more specialized journal.

What are the noteworthy results? The noteworthy results are that the authors have developed a mild method to convert epoxides into highly functionalized cyclopropanes via an oxygen-carbon exchange process.

Will the work be of significance to the field and related fields? While the overall transformation is quite interesting, this reviewer is unsure of the benefit of converting the carbonyl group to the epoxide vs directly to the olefin (which is formed in situ in the reaction sequence) for cyclopropanation. In cases where the epoxide already exists, this method would be great, however, this is likely more niche in nature.

How does it compare to the established literature? If the work is not original, please provide relevant references. This method is unique in that it is able to convert the epoxide in situ to the olefin. The established literature would go directly through cyclopropanation of the olefin.

Does the work support the conclusions and claims, or is additional evidence needed? Yes, the work supports the conclusions and claims.

Are there any flaws in the data analysis, interpretation and conclusions? Do these prohibit publication or require revision? Minor revisions noted below.

Is the methodology sound? Yes

Does the work meet the expected standards in your field? Yes

Is there enough detail provided in the methods for the work to be reproduced? Yes

Please find some additional comments for your consideration.

Page 1, Line 19-21 "One of the most important reactions of epoxides is their direct conversion to other three-membered (hetero)cyclic compounds such as thiiranes^{9–15}, aziridines^{16–20}, and cyclopropanes". This statement seems a bit strong as the conversion of epoxides to other three membered does not seem immediately obvious as one of the most importance reactions of epoxides. Consider using alternative language.

Figure 1. For the Hong and Cho methodology the "Bin" in the structure should be changed to "BPin".

Table 1. The authors should indicate in the text/table how the yields were determined when not isolated.

Table 1. For reaction optimization was each reaction condition tested more than once? Is there much batch variability?

Table1. When exploring the effect of temperature (40 –° 80 C) do the authors believe that this temperature was reached inside the reaction vessel given the boiling point of DCM is 40 °C? Was sufficient vapour pressure achieved to reach this temperature?

Table 1. Do the authors have an explanation for the different reactivity observed for the various ligands (TpBr₃, Tp(CF₃)₂, /Tp*) and/or sulfonyl hydrazones (2a,b,c)? If so, please include in the optimization text or include references to related work where appropriate.

Figure 2. In the text it is noted that example 16 employed modified reaction conditions, however, this is not noted in the Figure as it is for examples 19, 63, and 64.

Figure 2. For example 18, was any bis cyclopropanation observed? Do the authors think that cyclopropanation is occurring at the starting material olefinic site or the newly formed olefinic site (from the epoxide)? A labelling study would be interesting.

Figure 2. The X-ray structure image for compound 46 is not clear and a new image should be added.

Figure 2. The authors do a great job illustrating what substrates work well. What are the limitations of the methodology? From a medicinal chemistry perspective, it would be great to see additional azaheterocycles included (both as sulfonyl hydrazone or epoxide).

Figure 3. Please include reaction conditions in figure title.

Figure 3. The X-ray structure image for compound 66 is not clear and a new image should be added.

Figure 3. For cases when multiple olefinic sites exist is biscyclopropanation observed?

Figure 4. The image for T3' and T3 is not clear and a new image should be added.

Figure 5. The image for T3' and T3 is not clear and a new image should be added.

Comments for SI:

Typo in SI Title Page – Change "Refferences" to "References"

All purified compounds should include characterization data or be referenced to literature containing characterization (e.g. S2 on SI page 4). A statement indicating that the analytical data is in agreement with literature would also be good to include.

S2 on SI page 4, Line 92 – “bromopropyl” should be updated to “3-bromoprop-1-ene”

Consider shifting X-ray Crystallographic data after characterization data for the products.

SI page 17 – While it is great that the authors include a “Pre and Post reaction comparison” it would be easier to compare the Pre reaction if the reaction vessel was not submerged in the oil bath. Is it possible to include a picture of the reaction vessel before placement in the oil bath?

Point-by-point responses to editor and reviewer comments

Manuscript ID: NCOMMS-23-44730

Title: Late-stage skeletal transmutation of epoxides into trifluoromethyl cyclopropanes

Author(s): Linxuan Li, Paramasivam Sivaguru, Dandan Wei, Menglin Liu, Qingwen Zhu, Shuai Dong, Emanuele Casali, Nan Li, Giuseppe Zanoni and Xihe Bi*

Dear Reviewers

Thank you very much for your insightful comments and recommendations. We have carefully revised the manuscript in response to the reviewer's comments. The detailed corrections are included in the revised manuscript and are highlighted in yellow. The detailed revisions were listed as follows:

Reviewer 1:

Reviewer's general comment: The manuscript submitted by Professor Xihe Bi and co-workers presents a synthetic method of cyclopropanes with CF₃ group from epoxides. The developed reaction efficiently proceeds under mild conditions relative to the previous methods. The synthetic value of the unique transformation using Ag-carbene species was demonstrated by the derivatization of bioactive natural products and drug-like molecules with an array of functionalities.

The mechanistic analysis based on the DFT computation is also informative for potential readers. I, therefore, would like to recommend the publication of this paper in *Nature Communications*. I'd like to suggest some corrections and comments before publication.

Our response: We thank the referee for the positive comments and constructive suggestions on our work. These suggestions have helped us further improve the manuscript.

1) About the title, the developed method fundamentally provides the cyclopropanes possessing CF₃ group. Thus, a more concrete title might be appropriate, i.e., "Late-stage skeletal transmutation of epoxides into cyclopropanes with trifluoromethyl group".

Our response: We agree with the comment provided by the referee. Hence, we have changed the title to "Late-stage skeletal transmutation of epoxides into trifluoromethyl cyclopropanes".

2) In Figure 2, please spell out the "PCP" if it does not.

Our response: "PCP" is represented as "4-chlorophenyl" in Figure 2.

3) In Figure 4, regarding the reaction mechanism based on the DFT computation, I think that the most impressive elementary reaction is the concerted process from **Int2** into **Int3** via **TS2**. It would be appreciated if the authors also consider the stepwise pathway, like the attachments file.

Our response: Many thanks for the valuable suggestion. We have performed additional

computations to validate the possible deoxygenation path suggested by the reviewers. First, we considered the proposed mechanism depicted in Route 1 from the common oxonium ylide intermediate **Int2** at the same theoretical level. Our calculation results suggest that the energy barrier for the direct dissociation of **Int2** from the silver catalyst to form **Int2-1** is 61.1 kcal mol⁻¹, ruling out this route. Consider the possible pathway suggested by the reviewer (Route 2), which involves the tandem ring opening of **Int2** to form the carbocation intermediate **Int2-2** followed by the carbocation attack on the carbene carbon of **Int2-2** (i.e., tandem ring-opening cyclization) with the release of the silver catalyst to form the oxocyclic butane intermediate **Int2-2'**. However, our computational results could not find a stable **Int2-2** structure and thus excluded it. We also attempted to get **Int2-2'** directly from **Int2** through ring expansion (Route 3), but we were unable to compute the optimal transition state.

It should be noted that examining the reports on the mechanism of the deoxidation reaction of carbenes and epoxides shows that there is no evidence for the formation of the oxygen heterocyclic butane intermediate, and thus suggested route by the referee is likely less favored (see: *J. Am. Chem. Soc.* **1998**, *120*, 8681-8691; *Chem. Eur. J.* **2000**, *6*, 3777-3787; *Chem. Phys. Lett.* **2000**, *320*, 475-480).

Reviewer 2:

Reviewer's general comment: This paper describes a new cyclopropanation method via deoxygenation of epoxides under relatively mild conditions. With the careful choice of silver catalysts and sulfonylhydrazones as the carbene precursor, the direct transformation of a range of substituted epoxides was successfully achieved with favorable diastereoselectivity. Some control experiments along with DFT calculations provide mechanistic insights into the consecutive process which involves generation of alkene intermediates and subsequent [2+1] cycloaddition. Regarding the possible formation of ketones as the side product in the deoxygenation step, it's advisable to confirm the yield of trifluoromethyl ketone in the initial screening experiments of Table 1. Although other experimental results including synthetic applications are adequately provided, I am concerned about some weaknesses of this manuscript; (1) a limited scope of hydrazones accessible mainly from aromatic trifluoromethyl ketones, as well as (2) unachieved enantioselective applications. In sum, this is an excellent synthetic method contribution within its specific, demonstrated scope, but I feel it is just one step short of being published in a prestigious journal, *Nat. Commun.*

Response: We gratefully thank the reviewer for the excellent comments, criticisms, and suggestions, which have allowed us to modify the manuscript and provide new details to the work reported.

According to the reviewer's first concern, we have added the yield of trifluoroacetophenone **3'** to Table 1 and also revised the related description in the revised manuscript:

"At the onset of our study, our investigations focused on the reaction of 2-benzyloxirane (**1a**) and 2-trifluoromethylphenyl *N*-triflylhydrazone (**2a**, 2.0 equiv). A reaction system composed of catalytic $\text{Tp}^{\text{Br}3}\text{Ag}(\text{thf})$ and NaH (2.5 equiv) at 60 °C in dichloromethane (DCM) provided the desired cyclopropane **3** in 58% yield and 10:1 diastereomeric ratio (d.r.) along with 90% carbene deoxygenation product, trifluoroacetophenone **3'**, without the formation of competing C–O or C–H bond insertion products^{35,42–48}. Control experiments revealed that the amounts of **2a** and NaH were crucial to improving the reaction efficiency, and the best result (92% isolated yield of **3** and quantitative trifluoroacetophenone **3'**) was indeed obtained with 4.0 equiv of **2a** and 4.5 equiv of NaH (entries 1, 2 vs 3). Other silver catalysts, such as AgOTf, AgOAc, $\text{Tp}^{(\text{CF}_3)_2}\text{Ag}(\text{thf})$, and Tp^*Ag , resulted in diminished yields and diastereoselectivity (entries 4–7). This unusual reactivity of $\text{Tp}^{\text{Br}3}\text{Ag}(\text{thf})$ is mainly due to the presence of a substantial steric hindrance scorpion ligand ($\text{Tp}^{\text{Br}3}$), which not only hinders the occurrence of a competitive carbene dimerization process but also enhances non-enantioselectivity. To our surprise, no conversion of **1a** was observed when using $\text{Tp}^{\text{Br}3}\text{Cu}(\text{MeCN})$, $\text{Pd}(\text{OAc})_2$, and FeTPPCl as catalysts (entries 8–10). The use of $\text{Rh}_2(\text{OAc})_4$ as catalysts results in modest efficiency in the deoxidation process, thus obtaining **3** in poor yield and selectivity (entry 11). The reaction yield and stereoselectivity could not be improved by increasing (80 °C) or decreasing (40 °C) the reaction temperature (entries 12 and 13). The

substituents and their positions on *N*-sulfonylhydrazones were also critical to the reaction, as lower yields were observed with 4-methylphenyl- and 2,4,6-triisopropylphenyl *N*-sulfonylhydrazones (**2b** and **2c**) (entries 14 and 15). Note that the easy decomposition of *o*-trifluoromethyl *N*-triflylhydrazone compared to other functional group substituted *N*-sulfonylhydrazone enabled the rapid deoxidation process³⁵."

Entry	2 (equiv.)	Cat. (5 mol%)	NaH (equiv.)	Yield of 3' (%)	Yield of 3 (%)	d.r. of 3
1	2a (2.0)	Tp ^{Br3} Ag(thf)	2.5	90	58	10 : 1
2	2a (3.0)	Tp ^{Br3} Ag(thf)	3.5	99	82	10 : 1
3	2a (4.0)	Tp^{Br3}Ag(thf)	4.5	99	95 (92)^d	10 : 1
4	2a (4.0)	AgOTf	4.5	30	12	5 : 1
5	2a (4.0)	AgOAc	4.5	43	32	7 : 1
6	2a (4.0)	Tp ^{(CF3)2} Ag(thf)	4.5	82	60	7 : 1
7	2a (4.0)	Tp*Ag	4.5	60	22	7 : 1
8	2a (4.0)	Tp ^{Br3} Cu(MeCN)	4.5	56	0	-
9	2a (4.0)	Pd(OAc) ₂	4.5	0	0	-
10	2a (4.0)	FeTPPCL	4.5	8	trace	-
11	2a (4.0)	Rh ₂ (OAc) ₄	4.5	60	10	4 : 1
12 ^b	2a (4.0)	Tp ^{Br3} Ag(thf)	4.5	89	28	10 : 1
13 ^c	2a (4.0)	Tp ^{Br3} Ag(thf)	4.5	95	76	9 : 1
14	2b (4.0)	Tp ^{Br3} Ag(thf)	4.5	67	37	10 : 1
15	2c (4.0)	Tp ^{Br3} Ag(thf)	4.5	62	33	8 : 1

Ligands

Sulfonylhydrazones

To address the reviewer's second concern about the scope of hydrazones, we extended the scope of the reaction using various vinyl- and 1,3-dienyl-substituted *N*-triflylhydrazones as carbene precursors, which resulted in the corresponding products **59-62** in good yields and diastereoselectivity. Corresponding data and descriptions have also been added to the revised manuscript and are given below.

"*N*-Triflylhydrazones derived from trifluoromethyl vinyl/1,3-dienyl ketones were also found to be suitable carbene precursors, affording the corresponding alkenyl/dienyl substituted cyclopropanes **59-62** in high yields (75%-89%) with excellent diastereoselectivity. Interestingly, *N*-triflylhydrazone derived from pentafluoroethyl phenyl ketone was successfully converted into pentafluoroethyl cyclopropane **63** in 74% yield with a d.r. of >20:1, demonstrating the capability of the present work in divergent synthesis. This protocol also successfully extended to *N*-triflylhydrazones derived from ethyl phenylglyoxylate with high yield and selectivity (**64**)."

(See pages 78, 79, and 81 of Supplementary Information for NMR spectra of compounds 59, 60, and 62)

We also attempted to use alkyl *N*-trifosylhydrazones as carbene precursors in the reaction, but we did not find any cyclopropane products **79** and **81** instead producing competitive β -hydrogen elimination product **80** in 90% yield with 1,1,1-trifluoro-5-phenylpentan-2-one derived *N*-trifosylhydrazone (**2e**) under silver catalysis (Reaction a and b). Further screening of other catalysts revealed that the use of Rh₂(esp)₂ as catalysts could inhibit the competitive β -hydrogen elimination process, while they were ineffective in the deoxygenation process, likely due to the low reactivity of alkyl trifluoromethyl carbene intermediates. These results were added to the Supplementary Information (see page 18).

In response to address the reviewer's third concern about unrealized enantioselective applications, we tried to realize an enantioselective version of the reaction using chiral rhodium catalyst, Rh₂(*S*-PTAD)₄, which afforded the corresponding chiral cyclopropane product **46** in 54% yield with 88% ee. However, the incompatibility of alkyl-substituted carbenes in the reactions under rhodium-catalyzed conditions is a significant limitation of the process.

Chiralcel OJ-H, 0 % ⁱPr-OH in hexane, 0.8 mL/min $\lambda = 210$ nm, RT = 9.566 min, minor; RT = 14.531 min, major.

Racemate:

Signal: VWD1A, Wavelength=210 nm

RT [min]	Type	Width [min]	Area	Height	Area%
9.375	BM m	3.30	9613.39	262.75	50.26
14.651	BM m	6.39	9512.83	125.90	49.74
Sum			19126.22		

Enantiomer:

Signal: VWD1A, Wavelength=210 nm

RT [min]	Type	Width [min]	Area	Height	Area%
9.566	MM m	1.88	1813.58	48.05	5.73
14.531	MM m	6.50	29859.36	409.30	94.27
Sum			31672.94		

Minor issues:

(1) To corroborate the cyclopropanation pathway through the free alkene intermediates, it would be better to check a complete loss of optical purity in the reaction with nonracemic epoxides.

Our response: We sincerely appreciate the reviewer's suggestion. According to this suggestion,

we conducted a model reaction between (*R*)-styrene oxide **1b'** and *N*-trifosylhydrazone **2a**, which afforded the cyclopropane product **46** in 76% yield with the complete loss of optical activity was observed, suggesting the reaction proceeded through free alkene intermediate. The corresponding result has been added to the Supplementary Information (see, page 18) and given below.

Racemate:

Signal: VWD1A, Wavelength=210 nm

RT [min]	Type	Width [min]	Area	Height	Area%
9.375	BM m	3.30	9613.39	262.75	50.26
14.651	BM m	6.39	9512.83	125.90	49.74
Sum			19126.22		

Enantiomer:

Signal: VWD1A, Wavelength=210 nm

RT [min]	Type	Width [min]	Area	Height	Area%
9.406	BM m	3.26	11910.30	364.74	50.50
14.758	BM m	5.52	11672.97	167.55	49.50
Sum			23583.26		

(2) For the products **16** and **19** containing two cyclopropane units, relative stereochemical ratios other than the diastereomeric ratios due to 1,2-substituents on the cyclopropane need to be

specified.

Our response: We thank the reviewer for pointing out this error. Since the structure of compound **19** is symmetrical with only one pair of diastereomers, the current statement is correct. However, 1,2-substituents in compound **16** result in other stereochemical configurations. After careful analysis of the product structure of compound **16** by NOE spectra, we identified and added the stereochemical ratio of product **16** (d.r. >20:1:1:1) to Figure 2.

(3) In Figure 4B, I advise to renumber the product **3a**, trifluoromethyl acetophenone, which may confound with the cyclopropane product **3**.

Our response: Thank you for the reviewer's suggestion. We have revised the cyclopropane product to **3** and trifluoroacetophenone to **3'** in Table 1.

(4) Throughout the Supplementary Information; “trimethylsulfoxide iodide” should be replaced with “trimethylsulfoxonium iodide”.

Our response: The “trimethylsulfoxide iodide” has been corrected to “trimethylsulfoxonium iodide” in the Supplementary Information.

(5) Typos in the Supplementary Information:

P. 15, L .7; hydrotris(3,5-dimethylpyrazol-1-yl)borate.

P. 18; DFT Calculations.

Our response: We have corrected these typos in Supplementary Information.

Reviewer 3:

Reviewer's general comment: In this manuscript by Bi and coworkers, the transformation of epoxides into cyclopropanes in one pot by the action of *N*-triflylhydrazones as carbene precursors (and deoxygenation reagents) and a silver catalyst under basic conditions is reported. The authors demonstrate a remarkable substrate scope and functional group tolerance. The process performs equally well for some commercial drugs and other highly functionalized substrates. Based on control experiments and DFT computations, a reasonable mechanism is proposed. It consists of sequential epoxide deoxygenation with oxygen capture by the silver-carbene and cyclopropanation of the olefine intermediates to yield the products in good stereoselectivities. The entire process is termed as “skeletal editing”, a field of high topical interest. Overall, the described protocol grants access to a wide range of valuable products. However, at a second look, it is merely a one-pot process of epoxide deoxygenation (well-known) and olefin-cyclopropanation (well-known). This fact becomes further obvious because substrates containing terminal or disubstituted olefins do not appear in the substrate table. This shortcoming becomes apparent after consideration of the mechanism, but the authors failed to discuss this at an earlier stage. Thus, I am missing the conceptual novelty of the process if it would correspond to a proper “single step” skeletal editing, and I am less enthusiastic about recommending the manuscript for publication in *Nat. Comm.* Indeed, I would expect a less impactful reception of the same work if the terminology “one-pot” or “tandem,” more realistically reflecting the mechanism, would have been chosen. I am not an expert in organic synthesis, so I am unsure if this criticism is also mirrored in the broader community.

Response: We thank reviewer 3 for their constructive comments and thorough examination of the manuscript. These comments and suggestions have helped us further improve the manuscript. As suggested by the reviewer, we used the terminology “tandem reaction” or “sequential reaction” throughout the manuscript to highlight the importance of the present work.

Beyond that, the manuscript is well written, and the synthesis, analysis, and computational work is performed at a high level. The following points should be considered in any case:

Response: It is a great honor to receive your recognition for the writing, analysis, synthesis, and computational study of this work. Here is our response to your suggestions.

1) A more precise association of the processes in Figure 1 with the text, either by names or a)b) etc, would be helpful.

Our response: Thank you for the reviewer's suggestions. In accordance with the reviewer's recommendation, we have modified Figure 1 in the revised manuscript, which differentiates the association of four different reactions.

2) In substrate 8, Figure 2, why PMP?

Our response: Many thanks for this comment. The polarity of product **8** obtained from the reaction of epoxide with 4-chlorophenyl *N*-trifosylhydrazones and impurities are almost the same, which makes it difficult to obtain pure product. The use of 4-methoxyphenyl *N*-trifosylhydrazone as a substrate allowed the product polarity to be distinguished from impurities, allowing the pure product to be isolated.

3) In the discussion of the substrate scope, please comment on aliphatic carbene precursors.

Our response: Thanks for your comment. As per the suggestions, we attempted to use alkyl *N*-trifosylhydrazones as carbene precursors in the reaction, but we did not find any cyclopropane products **79** and **81** instead producing competitive β -hydrogen elimination product **80** in 90% yield with 1,1,1-trifluoro-5-phenylpentan-2-one derived *N*-trifosylhydrazone (**2e**) under silver catalysis (Reaction a and b). Further screening of other catalysts revealed that the use of $\text{Rh}_2(\text{esp})_2$ as catalysts could inhibit the competitive β -hydrogen elimination process, while they were ineffective in the deoxygenation process, likely due to the low reactivity of alkyl trifluoromethyl carbene intermediates. These results were added to the Supplementary Information (see page 18).

4) Page 6, line 133: not “alkaline” but rather “basic” conditions.

Our response: Corrected.

5) Page 7, line 164: “thermodynamically and kinetically more stable” should be

“thermodynamically and kinetically favored”.

Our response: Corrected.

6) Page 7, line 167: Please refer to Figure 5.

Our response: We have corrected it.

7) Improve the resolution of Figures 4 and 5.

Our response: Many thanks for this comment. The resolution of Figures 4 and 5 improved as follows:

Reviewer 4:

Reviewer's general comment: The manuscript by Li and coauthors describes a silver-catalyzed conversion of epoxides into cyclopropanes using *N*-trifosylhydrazones as carbene precursors via an oxygen-carbon exchange process. The reaction proceeds through initial conversion of the epoxide into the olefin, followed by subsequent cyclopropanation of the olefin. Mechanistic experiments, along with DFT calculations, support a reaction pathway involving silver-coordinated ylide fragmentation, followed by a [2+1] cycloaddition.

Overall, the manuscript is well written, with aesthetically pleasing figures. The substrate scope was shown to be quite broad, with good functional group tolerance. However, from a medicinal chemistry perspective, additional azaheterocycles would be desired.

While the overall transformation is quite interesting, this reviewer is unsure of the benefit of converting the carbonyl group to the epoxide vs directly to the olefin (which is formed in situ in the reaction sequence) for cyclopropanation. In cases where the epoxide already exists, this method would be great, however, this is likely more niche in nature. For this reason, this reviewer suggests publication in a more specialized journal.

What are the noteworthy results? The noteworthy results are that the authors have developed a mild method to convert epoxides into highly functionalized cyclopropanes via an oxygen-carbon exchange process.

Will the work be of significance to the field and related fields? While the overall transformation is quite interesting, this reviewer is unsure of the benefit of converting the carbonyl group to the epoxide vs directly to the olefin (which is formed in situ in the reaction sequence) for cyclopropanation. In cases where the epoxide already exists, this method would be great, however, this is likely more niche in nature.

How does it compare to the established literature? If the work is not original, please provide relevant references. This method is unique in that it is able to convert the epoxide in situ to the olefin. The established literature would go directly through cyclopropanation of the olefin.

Does the work support the conclusions and claims, or is additional evidence needed? Yes, the work supports the conclusions and claims.

Are there any flaws in the data analysis, interpretation and conclusions? Do these prohibit publication or require revision? Minor revisions noted below.

Is the methodology sound? Yes

Does the work meet the expected standards in your field? Yes

Is there enough detail provided in the methods for the work to be reproduced? Yes

Our response: We are grateful to the reviewer for providing a critical report and highlighting the

values of the present work. We receive all these comments as feedback and use them to further improve our manuscript quality.

Please find some additional comments for your consideration.

Page 1, Line 19-21 “One of the most important reactions of epoxides is their direct conversion to other three-membered (hetero)cyclic compounds such as thiiranes⁹⁻¹⁵, aziridines¹⁶⁻²⁰, and cyclopropanes”. This statement seems a bit strong as the conversion of epoxides to other three membered does not seem immediately obvious as one of the most importance reactions of epoxides. Consider using alternative language.

Our response: Many thanks for the reviewer's comment. We have rephrased the mentioned statement as follows:

"Epoxides readily transformed into other three-membered (hetero)cyclic compounds such as thiiranes⁹⁻¹⁵, aziridines¹⁶⁻²⁰, and cyclopropanes without altering the ring structure."

Figure 1. For the Hong and Cho methodology the “Bin” in the structure should be changed to “Bpin”.

Our response: We have corrected it.

Table 1. The authors should indicate in the text/table how the yields were determined when not isolated.

Our response: Thanks for your suggestions. We have added the following notes to Table 1:

"The yields and the diastereomeric ratio (d.r.) were determined by relative integration of ¹⁹F NMR spectra."

Table 1. For reaction optimization was each reaction condition tested more than once? Is there much batch variability?

Our response: Thanks. Yes. Each reaction condition was repeated at least three or more times during the optimization process, and the resulting error was less than 5%.

Table1. When exploring the effect of temperature (40–80 °C) do the authors believe that this temperature was reached inside the reaction vessel given the boiling point of DCM is 40 °C? Was sufficient vapour pressure achieved to reach this temperature?

Our response: Thanks for the valuable comment. Yes. We believe the reaction temperature (ie., inside the reaction vessel) has reached the set temperature of 40 °C, 60 °C, or 80 °C. The reaction tube we utilize is a high-pressure, thick-walled glass tube capable of achieving sufficient saturated vapor pressure to reach the required temperature.

Table 1. Do the authors have an explanation for the different reactivity observed for the various ligands (Tp^{Br3}, Tp^{(CF3)2}, /Tp*) and/or sulfonyl hydrazones (**2a,b,c**)? If so, please include in the optimization text or include references to related work where appropriate.

Our response: According to the reviewer's suggestions, we have added the following descriptions to the reaction conditions optimization section.

"This unusual reactivity of $\text{Tp}^{\text{Br}_3}\text{Ag}(\text{thf})$ is mainly due to the presence of a substantial steric hindrance scorpion ligand (Tp^{Br_3}), which not only hinders the occurrence of a competitive carbene dimerization process but also enhances non-enantioselectivity."

"Note that the easy decomposition of *o*-trifluoromethyl *N*-triftosylhydrazone compared to other functional group substituted *N*-sulfonylhydrazones enabled the rapid deoxidation process."

Figure 2. In the text it is noted that example **16** employed modified reaction conditions, however, this is not noted in the Figure as it is for examples **19**, **63**, and **64**.

Our response: Many thanks. We obtained compound **16** after prolonging the reaction time to 24 h, and the corresponding annotation was added to the title of Figure 2.

"^b Extend reaction time to 24 h."

Figure 2. For example **18**, was any bis cyclopropanation observed? Do the authors think that cyclopropanation is occurring at the starting material olefinic site or the newly formed olefinic site (from the epoxide)? A labelling study would be interesting.

Our response: Many thanks. Compounds **18** and **19** are derived from the same substrate, 1,2,7,8-diepoxyoctane, but yield different products based on the doses of *N*-sulfonylhydrazone used.

Under standard conditions, 1,4-di(oxiran-2-yl)butane readily converted to *n*-hexene-substituted cyclopropane **18** (80% yield and >12:1 d.r.) but produced the expected 1,4-dicyclopropylbutane **19** (74% yield and 8:1 d.r.) with 6 equiv. of 4-chlorophenyl trifluoromethyl *N*-sulfonylhydrazone.

Figure 2. The X-ray structure image for compound **46** is not clear and a new image should be added.

Our response: A new X-ray structure for compound **46** has been provided.

Figure 2. The authors do a great job illustrating what substrates work well. What are the limitations of the methodology? From a medicinal chemistry perspective, it would be great to see additional azaheterocycles included (both as sulfonyl hydrazone or epoxide).

Our response: We thank the reviewer for their useful comment. The limitation of our method is that it cannot target more complex natural products or drug molecules containing epoxy structures. The reaction may not proceed smoothly with more complex settings due to the presence of

sensitive groups or increased steric hindrance. In fact, we attempted to use drugs bearing epoxy structures, such as eplerenone and scopolamine but observed poor results. The selectivity of [2+1] cycloaddition to complex molecules containing alkenyl groups after deoxidation is also a notable limitation.

Figure 3. Please include reaction conditions in figure title.

Our response: We have added the reaction conditions to the Figure 3 caption.

"Figure 3. Late-stage skeletal editing to access trifluoromethyl cyclopropane containing quaternary carbon centers. Reaction conditions: ^a **1** (0.15 mmol, 1.0 equiv.), **2** (0.6 mmol, 4.0 equiv.), NaH (0.68 mmol, 4.5 equiv.) and Tp^{Br3}Ag(thf) (10 mol%) in DCM (4.0 mL) at 60 °C for 2-10 h. ^b **1** (0.15 mmol, 1.0 equiv.), **2** (0.38 mmol, 2.5 equiv.), NaH (0.38 mmol, 2.5 equiv.) and Tp^{Br3}Ag(thf) (5 mol%) in DCM (4.0 mL) at 60 °C for 2 h. Yields of the isolated product are given. Diastereomer ratio (d.r.) determination by relative integration of ¹⁹F NMR spectra. PCP, 4-chlorophenyl. PMP, 4-methoxyphenyl."

Figure 3. The X-ray structure image for compound **66** is not clear and a new image should be added.

Our response: A new X-ray structure for compound **69** (**66** in the previous manuscript) has been provided.

Figure 3. For cases when multiple olefinic sites exist is biscyclopropanation observed?

Our response: We thank the reviewer for their valuable comments. When there were multiple reaction sites, we did not monitor the product of bicyclic propane. Because aryl olefins are more reactive than alkyl olefins, carbenes preferentially undergo [2+1] cycloaddition with higher concentrations of aryl olefins formed in a short time for compounds **74** and **75** (see control experiments in Figure 4, the silver carbene deoxidation process is rapid). For compound **68**, the intermediate obtained in the reaction contains two olefins, where the carbene preferentially undergoes cycloaddition at the least substituted olefin.

Figure 4. The image for **T3'** and **T3** is not clear and a new image should be added.

Our response: Many thanks. As suggested by the reviewer, we have added the clear new images for **TS3** and **TS3'** to Figure 4.

Figure 5. The image for **T3'** and **T3** is not clear and a new image should be added.

Our response: Many thanks. As suggested by the reviewer, we have added the clear new images for **TS3** and **TS3'** to Figure 5.

Comments for SI:

Typo in SI Title Page – Change “Refferences” to “References”

Our response: We have corrected it.

All purified compounds should include characterization data or be referenced to literature containing characterization (e.g. S2 on SI page 4). A statement indicating that the analytical data is in agreement with literature would also be good to include.

Our response: We have corrected it.

S2 on SI page 4, Line 92 – “bromopropyl” should be updated to “3-bromoprop-1-ene”

Our response: We have corrected it.

Consider shifting X-ray Crystallographic data after characterization data for the products.

Our response: Many thanks. We have corrected it.

SI page 17 – While it is great that the authors include a “Pre and Post reaction comparison” it would be easier to compare the Pre reaction if the reaction vessel was not submerged in the oil bath. Is it possible to include a picture of the reaction vessel before placement in the oil bath?

Our response: Thank you for the reviewer's comments. We repeated the gram-scale experiment and obtained almost identical results as before. As per the suggestions, we have included a picture of the reaction vessel before placement in the oil bath as well as the pre and post reaction comparison on page 17 of the Supplementary Information.

In addition to the above revisions, we have also made the following changes to the revised manuscript

1) As shown in Figure 4, we have modified the catalytic cycle to make it clear and easier to understand.

2) The overall layout of the manuscript has been altered to correspond to the submission

format.

3) The chemical structures were formatted using the ChemDraw template provided in the guide.

4) The description of the method has been added to the manuscript, as is the need:

Methods

General procedure for the synthesis of fluoroalkyl *N*-triftosylhydrazones. Under air conditions, fluoroalkyl ketone (11 mmol, 1.1 equiv) and 20.0 mL EtOAc were added in an oven-dried 50.0 mL round-bottomed flask and then TfsNHNH₂ (10 mmol, 1.0 equiv) was added and the reaction mixture was stirred at 40 °C. Upon complete dissolution, 2.0 mL of boron trifluoride ether (48 wt% BF₃) was added. After complete consumption of the starting material (as evidenced by TLC), saturated sodium chloride solution was added and then extracted with EtOAc (20.0 mL x 3). The combined organic layers were dried over anhydrous Na₂SO₄ and concentrated under reduced pressure. The crude product was purified by silica gel column chromatography (PE/EtOAc = 10/1) to obtain the product with a white solid.

General procedure oxygen-carbon exchange reaction of alkyl substituted epoxides. In the glove box, *N*-sulfonylhydrazones derived from aryl trifluoromethyl ketone (0.60 mmol, 4.0 equiv), NaH (0.68 mmol, 4.5 equiv), and 4.0 mL of DCM were added to the dry sealed tube. Then the alkyl-substituted epoxides (0.15 mmol) and Tp^{Br₃}Ag(thf) (10 mol%) were added under stirring conditions. The resulting mixture was sealed and heated to 60 °C. After the reaction was completed and cooled to room temperature, the reaction mixture was filtered through diatomaceous earth under reduced pressure and the filter pad was washed with DCM (5.0 mL x 3). The combined residue was then concentrated under reduced pressure and the crude residue was purified by flash silica gel column chromatography to give cyclopropanation products.

General procedure oxygen-carbon exchange reaction of aryl substituted epoxides. In the glove box, *N*-sulfonylhydrazones derived from aryl trifluoromethyl ketone (0.38 mmol, 2.5 equiv.), NaH (0.38 mmol, 2.5 equiv.), and 4.0 mL of DCM were added to the dry-sealed tube. Then the aryl-substituted epoxides (0.15 mmol) and Tp^{Br₃}Ag(thf) (2.5-5 mol%) were added under stirring conditions. The resulting mixture was sealed and heated to 60 °C. After the reaction was completed and cooled to room temperature, the reaction mixture was filtered through diatomaceous earth under reduced pressure and the filter pad was washed with DCM (5.0 mL x 3). The combined residue was then concentrated under reduced pressure and the crude residue was purified by flash silica gel column chromatography to give cyclopropanation products.

General procedure oxygen-carbon exchange reaction of *N*-triftosylhydrazones derived from aryl aldehyde. In the glove box, *N*-triftosylhydrazones derived from aryl aldehyde (0.45 mmol, 3.0 equiv.), NaH (0.45 mmol, 3.0 equiv.), and 4.0 mL of DCM were added to the dry-sealed tube. Then the aryl-substituted epoxides (0.15 mmol) and Tp^{Br₃}Ag(thf) (5 mol%) were added under

stirring conditions. The resulting mixture was sealed and heated to 60 °C. After the reaction was completed and cooled to room temperature, the reaction mixture was filtered through diatomaceous earth under reduced pressure and the filter pad was washed with DCM (5.0 mL × 3). The combined residue was then concentrated under reduced pressure and the crude residue was purified by flash silica gel column chromatography to give cyclopropanation products.

5) We have unified the terminology used for "epoxide" and "oxirane" throughout the manuscript.

6) We have added the following annotations to the captions of Table 1, Figure 2, and Figure 3 to explain the source of the diastereomeric ratio.

"Diastereomer ratio (d.r.) determination by relative integration of ^{19}F NMR spectra."

Finally, we would like to show our great respect to all Reviewers, whose critical reviews and invaluable suggestions definitely have improved the quality of this manuscript. We hope that the revised manuscript will reach the level for publication in *Nature Communications*.

Thank you once again. We are looking forward to hearing from you.

Sincerely yours,

Xihe Bi

REVIEWERS' COMMENTS

Reviewer #1 (Remarks to the Author):

The manuscript submitted by Professor Xihe Bi and co-workers has been revised based on the reviewers' comments. The authors made considerable efforts to address the pointed-out issues. Consequently, my concerns have been resolved. Considering the revisions proposed by the other referees, I think that this work is ready for publication after the final proofreading. I would like to congratulate the authors for the job well done.

Reviewer #2 (Remarks to the Author):

The resubmitted paper has been adequately revised based on the reviewers' comments and I believe that the quality has been satisfactorily improved after incorporating additional experimental results. I additionally suggest replacing the nomenclatural inappropriate term "trifluoromethyl cyclopropanes" with "trifluoromethylcyclopropanes" throughout the manuscript.

Overall, I'm happy to support publication in Nature Communications.

Reviewer #3 (Remarks to the Author):

The authors did a good job of addressing the concerns of this reviewer. Technically, it's an excellent paper. Still unsure about the conceptual novelty and surprised that three tentative contra votes vs one pro vote lead to acceptance...

Reviewer #4 (Remarks to the Author):

This reviewer has read the revised manuscript "Late-stage skeletal transmutation of epoxides into trifluoromethyl cyclopropanes" by Professor Xihe Bi and coauthors and believes they have done a good job addressing/incorporating the reviewers questions/concerns.

Based on the revised manuscript, some additional considerations can be found below:

Given that the substrate scope also includes other functional groups (CF₂CF₃, CO₂Et, and H) than the CF₃ group, consider removing the specific mention of "trifluoromethyl" in the manuscript title. In addition, the use of "late-stage" also seems unnecessary as based off of the authors response (see below), general late-stage modification may be a challenge. Consider the title "A silver-catalyzed direct conversion of epoxides into cyclopropanes"

"We thank the reviewer for their useful comment. The limitation of our method is that it cannot target more complex natural products or drug molecules containing epoxy structures. The reaction may not proceed smoothly with more complex settings due to the presence of sensitive groups or increased steric hindrance. In fact, we attempted to use drugs bearing epoxy structures, such as eplerenone and

scopolamine but observed poor results. The selectivity of [2+1] cycloaddition to complex molecules containing alkenyl groups after deoxidation is also a notable limitation.”

Consider including the author names in Fig 1a when describing the various strategies employed to convert epoxides to cyclopropanes to help guide the reader through the figure.

Line 19-22: Consider changing “In light of their value, the transformation and utilization of readily accessible epoxides is of utmost importance in synthetic chemistry. Epoxides readily transformed into other three-membered (hetero)cyclic compounds such as thiiranes^{9–15}, aziridines^{16–20}, and cyclopropanes without altering the ring structure.” To “In light of their value, the transformation and utilization of readily accessible epoxides is of utmost importance in synthetic chemistry and includes the direct conversion into other three-membered (hetero)cyclic compounds such as thiiranes^{9–15}, aziridines^{16–20}, and cyclopropanes without altering the ring structure.”

Line 60-61: Consider changing “The reaction yield and stereoselectivity could not be improved by increasing (80 °C) or decreasing 60 (40 °C) the reaction temperature (entries 12 and 13).” to “The reaction yield and stereoselectivity could not be improved by decreasing (40 °C) or increasing (80 °C) the reaction temperature (entries 12 and 13).” to correspond with the table entries.

Consider including the enantioselective attempt using Rh₂(S-PTAD)₄ in the Supplementary Information.

Consider including some of the examples of limitations (e.g. azacycles, sterics) listed by the authors in their response in the Supplementary Information to help inform the reader. In the case of eplerenone and scopolamine, even low conversion is still useful when it allows access to novel chemical matter. It would also be good to make a reference in the body of manuscript to see the Supplementary Information for additional substrate information.

Point-by-point responses to editor and reviewer comments

Manuscript ID: NCOMMS-23-44730A

Title: Silver-catalyzed direct conversion of epoxides into cyclopropanes using *N*-triflylhydrazones

Author(s): Linxuan Li, Paramasivam Sivaguru, Dandan Wei, Menglin Liu, Qingwen Zhu, Shuai Dong, Emanuele Casali, Nan Li, Giuseppe Zanoni and Xihe Bi*

Thank you very much for considering our manuscript for publication. We are grateful to the editors and reviewers for their final technical and insightful comments and recommendations. We have carefully revised the manuscript in response to the reviewers' comments. The detailed corrections are included in the revised manuscript and are highlighted in yellow. A point-by-point response to the editor's and reviewer's comments is listed as follows:

Reviewer 1:

Reviewer's general comment: The manuscript submitted by Professor Xihe Bi and co-workers has been revised based on the reviewers' comments. The authors made considerable efforts to address the pointed-out issues. Consequently, my concerns have been resolved. Considering the revisions proposed by the other referees, I think that this work is ready for publication after the final proofreading. I would like to congratulate the authors for the job well done.

Response: We thank the reviewer for constructive comments and insightful advice that helped us improve the manuscript.

Reviewer 2:

Reviewer's general comment: The resubmitted paper has been adequately revised based on the reviewers' comments and I believe that the quality has been satisfactorily improved after incorporating additional experimental results. I additionally suggest replacing the nomenclatural inappropriate term "trifluoromethyl cyclopropanes" with "trifluoromethylcyclopropanes" throughout the manuscript.

Overall, I'm happy to support publication in *Nature Communications*.

Our response: We thank the reviewer for the excellent comments and insightful suggestions that helped us improve the manuscript. As per the suggestions, we have replaced the term "trifluoromethyl cyclopropanes" with "trifluoromethylcyclopropanes" throughout the manuscript.

As per the recommendations from another reviewer, we have changed the title as follows:

"Silver-catalyzed direct conversion of epoxides into cyclopropanes using *N*-triflylhydrazones"

Reviewer3:

Reviewer's general comment: The authors did a good job of addressing the concerns of this reviewer. Technically, it's an excellent paper. Still unsure about the conceptual novelty and surprised that three tentative contra votes vs one pro vote lead to acceptance...

Our response: We highly appreciate this reviewer for taking the time and effort to review our manuscript, and their excellent comments and insightful suggestions, especially on the novelty of the work, helped to significantly improve the quality of our manuscript.

We respect the reviewer's opinion. However, we are confident that the significance and the novelty of this work warrant its publication in *Nat. Commun.*

Reviewer 4:

Reviewer's general comment: This reviewer has read the revised manuscript "Late-stage skeletal transmutation of epoxides into trifluoromethyl cyclopropanes" by Professor Xihe Bi and coauthors and believes they have done a good job addressing/incorporating the reviewers questions/concerns.

Response: We thank the reviewer for constructive comments and insightful advice that helped us improve the manuscript.

Based on the revised manuscript, some additional considerations can be found below:

Given that the substrate scope also includes other functional groups (CF₂CF₃, CO₂Et, and H) than the CF₃ group, consider removing the specific mention of "trifluoromethyl" in the manuscript title. In addition, the use of "late-stage" also seems unnecessary as based off of the authors response (see below), general late-stage modification may be a challenge. Consider the title "A silver-catalyzed direct conversion of epoxides into cyclopropanes"

"We thank the reviewer for their useful comment. The limitation of our method is that it cannot target more complex natural products or drug molecules containing epoxy structures. The reaction may not proceed smoothly with more complex settings due to the presence of sensitive groups or increased steric hindrance. In fact, we attempted to use drugs bearing epoxy structures, such as eplerenone and scopolamine but observed poor results. The selectivity of [2+1] cycloaddition to complex molecules containing alkenyl groups after deoxidation is also a notable limitation."

Our response: Many thanks for this comment. We agree with the reviewer suggestions and, after comprehensive consideration, have changed the title as follows:

"Silver-catalyzed direct conversion of epoxides into cyclopropanes using *N*-trifosylhydrazones"

Consider including the author names in Fig 1a when describing the various strategies employed to convert epoxides to cyclopropanes to help guide the reader through the figure.

Our response: Many thanks for their valuable suggestions. We have included the author names in Fig. 1a. The modified version of Fig. 1a is given below:

Line 19-22: Consider changing “In light of their value, the transformation and utilization of readily accessible epoxides is of utmost importance in synthetic chemistry. Epoxides readily transformed into other three-membered (hetero)cyclic compounds such as thiiranes⁹⁻¹⁵, aziridines¹⁶⁻²⁰, and cyclopropanes without altering the ring structure.” To “In light of their value, the transformation and utilization of readily accessible epoxides is of utmost importance in synthetic chemistry and includes the direct conversion into other three-membered (hetero)cyclic compounds such as thiiranes⁹⁻¹⁵, aziridines¹⁶⁻²⁰, and cyclopropanes without altering the ring structure.”

Our response: Thank you for your valuable suggestion. We have already made the following changes to this sentence:

"In light of their value, the transformation and utilization of readily accessible epoxides is of utmost importance in synthetic chemistry and includes the direct conversion into other three-membered (hetero)cyclic compounds such as thiiranes⁹⁻¹⁵, aziridines¹⁶⁻²⁰, and cyclopropanes without altering the ring structure."

“In light of their value, the transformation and utilization of readily accessible epoxides is of utmost importance in synthetic chemistry and includes the direct conversion into other three-membered (hetero)cyclic compounds such as thiiranes⁹⁻¹⁵, aziridine^{s16-20}, and cyclopropanes without altering the ring structure.”

Line 60-61: Consider changing “The reaction yield and stereoselectivity could not be improved by increasing (80 °C) or decreasing 60 (40 °C) the reaction temperature (entries 12 and 13).” to “The reaction yield and stereoselectivity could not be improved by decreasing (40 °C) or increasing (80 °C) the reaction temperature (entries 12 and 13).” to correspond with the table entries.

Our response: Many thanks for the valuable suggestion. As per the suggestions, we have made the changes in the revised manuscript that correspond with the table entries.

Consider including the enantioselective attempt using $\text{Rh}_2(\text{S-PTAD})_4$ in the Supplementary Information.

Our response: As suggested by the reviewer, we included the enantioselective attempt with $\text{Rh}_2(\text{S-PTAD})_4$ in the Supplementary Information on page 19.

Consider including some of the examples of limitations (e.g. azacycles, sterics) listed by the authors in their response in the Supplementary Information to help inform the reader. In the case of eplerenone and scopolamine, even low conversion is still useful when it allows access to novel chemical matter. It would also be good to make a reference in the body of manuscript to see the Supplementary Information for additional substrate information.

Our response: According to the reviewer's suggestion, we have added unreactive substrates in the Supplementary Information on page 18. And referenced them in the body of the manuscript:

"We also attempted to use alkyl *N*-trifosylhydrazones as carbene precursors (For details, please see Supplementary Figure 1b)..."

"We attempted to use allose derived epoxides as substrates for the reaction because the steric hindrance of the trisubstituted olefin intermediates prevented further conversion of cyclopropane. The derivatives of scopolamine, due to their alkalinity, result in catalyst deactivation and failure to obtain the corresponding cyclopropane product (For details, please see Supplementary Figure 1b)."

In addition to the above revisions, we have added the necessary brief summary here:

"Author introduce a silver carbene strategy to achieve O-to-C atom exchange of epoxides in a single step using fluoroalkyl carbene derived from fluoroalkyl *N*-trifosylhydrazones, resulting in diverse fluoroalkylcyclopropanes."

Finally, we would like to show our great respect to all Referees, whose critical comments and invaluable suggestions have improved the quality of this manuscript.

Thank you once again for your efforts and consideration of this manuscript for *Nature Communications*. We are looking forward to hearing from you.

Sincerely yours,

Xihe Bi